# Embedding Recycling for Language Models

## Abstract

Training and inference with large neural models is expensive. However, for many application domains, while new tasks and models arise frequently, the underlying documents being modeled remain mostly unchanged. We study how to decrease computational cost in such settings through *embedding recycling (ER):* re-using activations from previous model runs during training or inference. In contrast to prior work focusing on freezing small classification heads for fine-tuning which often leads to notable drops in accuracy, we propose caching an intermediate layer's output from a pretrained model and fine-tuning the remaining layers for new tasks. We show that our method is effective using either fine-tuning for the trainable layers, or parameter-efficient adapters. For the best-performing model in our experiments, DeBERTa-v2 XL with adapters, we find that our method provides a 100% speedup during training and an 87-91% speedup for inference, and has negligible impacts on accuracy averaged across eight tasks spanning text classification and entity recognition in the scientific domain and general-domain question answering. Further, in experiments with SciBERT, BERT-base, and RoBERTa-large, we show a 100% speedup during training and a 55-86% speedup for inference, at only a 0.19-0.23% reduction in accuracy on average. Finally, we identify several open challenges and future directions for ER.

## 1 Introduction

Large pretrained language models form the foundation of modern NLP, and continue to push the state-of-the-art on a wide range of natural language processing tasks (Devlin et al., 2019; Liu et al., 2019b; Bommasani et al., 2021). Larger models tend to offer superior accuracies (Kaplan et al., 2020), but also higher computational costs. The steep computational cost associated with large neural language models slows down experimentation, increases financial barriers to the technology, and contributes to global climate change (Strubell et al., 2019; Dodge et al., 2022).

We explore how to reduce this computational cost using the simple observation that often, the text we wish to process with a large neural model has already been processed through a model before—and the activations from previous runs may be re-used to speed up the current one. In many corpora, substantial portions of the text remain relatively fixed over time (e.g. scientific papers, Wikipedia articles, StackExchange posts, financial reports, legal records, etc.). Further, a number of models for various tasks are run over this text (entity recognition, topic classification, relation extraction, summarization, question answering, and so on). Any one of the runs produces a contextualized embedding of the text; and even without task-specific fine-tuning, contextualized embeddings produced by pretrained language models are known to capture syntactic and semantic knowledge about their input texts (Goldberg, 2019; Wiedemann et al., 2019; Rogers et al., 2020), which can be useful for a variety of downstream tasks.

We study *embedding recycling* (ER), the technique of leveraging activations from previous model runs in order to improve the accuracy and efficiency of future training and inference. While previous work has explored a similar approach using frozen encoders (Du et al., 2020), it focuses on small classification heads for fine-tuning which lead to drops in accuracy. By contrast, we experiment with a simple *layer-recycling* ER method that stores a cache of the activations from an intermediate layer of a pretrained model, and then starts from those activations when the same input sequence is seen again during fine-tuning or inference. Layer recycling imposes a small additional time cost the first time a model is run on a text, in order to

compute and write the cache. Then, all subsequent runs on the text can start from the pretrained cache, which we show results in substantial increases in throughput for those runs at small or no cost to model accuracy. This is significant because the number of such subsequent runs can be large in some real-world applications, including when re-training models many times during development, executing many epochs in a training run, or performing inference with many different models over the same corpus Du et al. (2020); Wei et al. (2022).

To summarize, we make the following contributions:

- We propose embedding recycling as a method for lowering the computational costs of training and inference for language models, and explore two layer recycling methods, one that uses standard fine-tuning and another that uses parameter-efficient adapters.

- In experiments across seven models and eight tasks, we show that layer recycling is generally effective. For the best-performing model on our tasks, DeBERTa-XL with adapters, we find that layer recycling matches performance of the original model while providing a 87-91% speedup at inference time, and a 100% speedup at training time.

- We explore open challenges for embedding recycling and present questions for future work.

## 2 Related Work

Transformer-based pretrained language models (PLMs) are typically first pretrained over a large text corpus and then fine-tuned on a downstream task. Recent work analyzing the structures of neural networks has shown that different layers within transformer models extract different features from textual data. Studies using canonical correlation analysis have found that the shallower layers tend to converge earlier in training than the deeper layers (Raghu et al., 2017; Morcos et al., 2018). Kovaleva et al. (2019) observed that the last layers of BERT change the most substantially during fine-tuning, suggesting that earlier layers tend to extract universal features whereas later layers focus on task-specific modeling.

This tendency of deeper layers to learn task-specific knowledge has led to a variety of work finding that it is often not necessary to train all the layers of a model. Lee et al. (2019) found that only a fourth of the final layers in BERT and RoBERTa need to be fine-tuned to achieve 90% of the fully fine-tuned performance. Other work explored adaptive approaches that vary the number of frozen layers over the course of training, approaching or exceeding the performance of fully fine-tuned models while substantially speeding up the training process (Raghu et al., 2017; Xiao et al., 2019; Brock et al., 2017).

Similar to our work, some prior work on dynamic freezing also employed caching mechanisms to eliminate the cost of the forward pass for frozen layers (Liu et al., 2021; He et al., 2021). However, in that work the dynamic choice of how many layers to freeze means that the cached activations are only useful at training time and only for a single task; we propose caching embeddings from the pretrained model, which can then be reused across multiple downstream tasks and applied at inference time as well.

Other recent studies have sought to improve model inference speed by skipping computations in later layers. Sajjad et al. (2020) found that for some tasks and PLMs, up to half of the layers can be removed from the model to obtain a 98-100% speedup at the cost of a 1-3% drop in task performance. Kumar et al. (2019) considered approximate caching to skip the deeper layers for inputs that produce similar intermediate layer representations. Early exit strategies have also been proposed, which allow the model to dynamically decide when to skip later layers (Cambazoglu et al., 2010; Xin et al., 2020). In contrast, embedding recycling focuses on eliminating computation of earlier layers rather than later ones, which makes it potentially possible to combine embedding recycling with early exiting.

SkipBERT (Wang et al., 2022) combined early exiting with an approach in which cached n-gram embeddings (tri-grams in practice) were used to approximate the intermediate activations of new inputs. However, SkipBERT only measures latency (with a batch size of 1), targeting the use case where individual new inputs need to be processed quickly. In embedding recycling, we focus on the case where an entire cached corpus needs to be processed at once, making throughput more important than small-batch latency. In

addition, SkipBERT's approach is only evaluated on BERT-base and only on sentence-level tasks, whereas we demonstrate the generality of embedding recycling across 8 models and 3 task types.

Precomputing text representations to speed up future processing on the same data is commonly done when creating fixed-size document-level embeddings for use on document-level tasks (Conneau et al., 2017; Cohan et al., 2020); in contrast, we study contextualized *token-level* embeddings that can be used for tasks such as named entity recognition (NER) and question answering. ReadOnce Transformers (Lin et al., 2021) do consider multi-task variable-length document representations, but do so in the context of "representation+text" style tasks such as question answering, where a cached document representation is paired with a query text at inference time (such as a question or prompt); the approach is pretrained with question answering data and evaluated on QA and summarization, rather than tasks such as text classification or NER where the entire input can be cached.

Du et al. (2020) propose a similar high-level idea to ours in that they do cache general-purpose token-level model representations, trained in a multi-task setting; however, the approach in that work only applies a small MLP to the stored representations and reports a meaningful drop in accuracy (greater than 2% on average) compared to fully fine-tuned models. We find that reusing the later layer parameters of a pretrained transformer in addition to the cached activations of prior layers enables us to often essentially match fully fine-tuned model accuracy on average, while decreasing computational cost. As noted in our experiments, if we instead use small MLPs from a frozen pretrained encoder, we see large drops in accuracy on our tasks.

Wei et al. (2022) combine the practices of freezing shallow layers and knowledge distillation to create a multi-task model. However, they use a two stage process where $12-N$ deep layers are fine-tuned for each individual task keeping $N$ frozen layers. This is followed by distillation of the $N$ layers for further computational gains. We take advantage of the parameter efficient adapter modules (Houlsby et al., 2019), and replace this process with a single step of fine-tuning a frozen base model that has adapters attached only to the deeper layers.

Our work also has connections to work on memory- and retrieval-augmented language modeling. Prior work on using memory (e.g., Grave et al. (2016); Dai et al. (2019); Rae & Razavi (2020); Wu et al. (2022)) generally focuses on modeling long range context and caching representations of older history in a sequence, while work on retrieval (e.g., (Guu et al., 2020; Karpukhin et al., 2020)) focuses on fetching text from a knowledge base or corpus to serve as additional context. In both cases, the aim is to use representations of additional text (from earlier in a document or from a knowledge base) to improve modeling of new inputs. In contrast, our work focuses on caching the representations of an entire sequence to speed up computation for new tasks.

## 3 Methods

In the transformer architecture (Vaswani et al., 2017), an input sequence $x$ of length $S$ and dimension $d$ is transformed with a function $F : \mathbb{R}^{S \times d} \to \mathbb{R}^{S \times d}$ defined by the composition of $N$ transformer layers $F^{(1)}, ..., F^{(N)}$ as follows:

$$\texttt{F}^\ell(x) = \texttt{LN}(\texttt{FF}^\ell(\ x') + x') \tag{1}$$

$$\texttt{x'} = \texttt{LN}\Big(\texttt{MH}^\ell(x) + x\Big) \tag{2}$$

where $\texttt{LN}$ is a layer normalization (Ba et al., 2016), $\texttt{FF}$ is a feed forward network, and $\texttt{MH}$ is the self-attention layer that consists of multiple heads and contextualizes the input sequence vector. The output of each layer is used as input to the next layer.

$$h^{\ell+1} = F^\ell(h^\ell) \tag{3}$$

Our approach is to cache the output representations $h^k \in \mathbb{R}^{S \times d}$ at a certain layer $k$ and reuse them for fine-tuning on a new given task. We refer to this process of caching and reusing the output representations

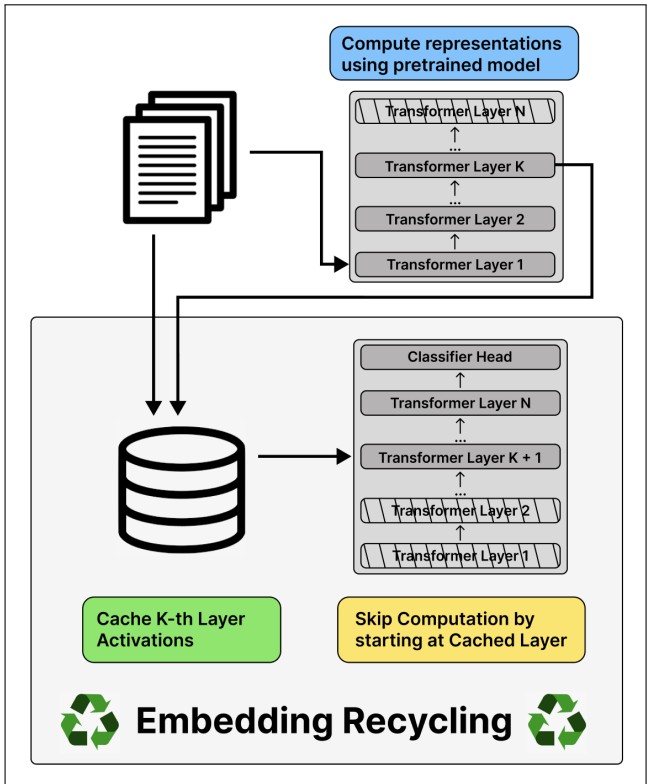

Figure 1: Overview of the embedding recycling approach. In the figure, the K-th layer activations are saved for future fine-tuning on downstream tasks, skipping redundant computations of earlier layers in the transformer model.

of a layer as *layer recycling*. This enables us to reduce the size of the transformer model from $N$ layers to $N - k$ layers, reducing the computational cost during fine-tuning and inference.

Note that the key requirement of layer recycling is that we first need to process the entire data with the transformer model and cache the representations, so that we could later reuse these representations many times during fine-tuning and inference on new tasks. We experiment with two types of layer recycling approaches as explained next.

We start with a pretrained transformer $F$ (e.g., BERT) consisting of $F^{(1)}, ..., F^{(k)}, ..., F^{(N)}$ layers. We run the transformer over a corpus $\mathcal{C}$ and cache the output representations of layer $k$ for each instance $c$ in $\mathcal{C}$, i.e., $h^k_{c \in \mathcal{C}}$. The same transformer model is then used for fine-tuning on new tasks, however, instead of fine-tuning all the layers, we only fine-tune the parameters of the latter $N - k$ layers $F^{(k+1)}, ..., F^{(N)}$. We can either train all of the weights in these layers (we refer to these as *reduced models*), or only train adapter modules added on the layers (discussed below). In either case, for the instance $c$ in the dataset $\mathcal{C}$ we simply retrieve and use the previously cached representation $h^k_{c \in \mathcal{C}}$ as input to layer $F^{(k+1)}$. This avoids the extra computation through layers $F^{(1)}, ..., F^{(k)}$ but adds a small cost for retrieving the representation from storage (see subsection 5.3 for efficiency analysis).

### 3.1 Adapters

In addition to the standard layer recycling, we evaluate whether combining layer recycling with Adapter modules (Houlsby et al., 2019) can improve performance as compared to fully fine-tuned models. Adapters are typically used to improve the parameter efficiency of finetuning and mitigate the storage costs of large language models. They also enable more sample-efficient fine-tuning and can result in improved fine-tuning

performance (Karimi Mahabadi et al., 2022). Therefore, we investigate whether Adapters can also yield accuracy improvements when used in combination with layer recycling.

Adapter modules contain a down-projection, an up-projection, and a residual connection module: $h \leftarrow h + (f(h\mathbf{W}_{down})\mathbf{W}_{up})$. The adapters are separately inserted after the MH and the FF layers in the transformer architecture (Equation 2). Further, Rücklé et al. (2021) experiment with dropping adapters from the lower transformer layers to provide inference time speedup. In our experiments, adapters are added to the latter half of transformer layers in the reduced transformer models. Particularly, as in the reduced models discussed earlier, with adapters the pretrained original transformer $F$ first caches the intermediate activations $h^k_{c \in \mathcal{C}}$ for each input in a selected corpus at layer $k$. Then the first $k$ layers are removed from the transformer. During fine-tuning, the cached representations are fed as input to the later $N - k$ layers of the transformer, which consist of the frozen transformer layers plus trainable adapter parameters. Thus, we fine-tune only the additional 6-8% parameters introduced by the adapters. We refer to learning adapters on all layers as the *full adapter* setting and the layer recycling version as the *reduced adapter* setting.

## 4 Experimental Setup

We now present our experiments evaluating whether recycled embeddings can be paired with reduced large language models to maintain accuracy while improving training and inference speed. We explore the effectiveness of embedding recycling across a variety of different tasks, datasets, and transformer models.

### 4.1 Models

Our full-size models include the encoder transformers BERT, SciBERT (Beltagy et al., 2019), RoBERTa (Liu et al., 2019b), and DeBERTa (He et al., 2020). We also experiment with the encoder-decoder T5 model (Raffel et al., 2019). We selected these architectures since they are widely-used pretrained transformers across a variety of tasks in different domains. We experiment with multiple sizes of these models, including distilled (Sanh et al., 2019; Wang et al., 2020; 2021), base, and large variants, to gauge the effectiveness of recycled embeddings with an increase in the network size.

To investigate the effectiveness of layer recycling, we test several reduced models in which we use caching to reduce 50% of the layers (e.g., caching layer 12 in RoBERTa-large and layer 6 in BERT-base).[1] We compare each reduced model to its fully fine-tuned counterpart across the text classification, NER, and QA tasks. The hardware details and hyperparameters for our models are specified in Appendix A.

### 4.2 Datasets

For our experiments, we focus on three core NLP tasks: text classification, named-entity recognition (NER), and extractive question-answering (QA). Scientific papers, due to their immutable nature, are an especially appropriate target for embedding recycling, so we focus much of our evaluation on the scientific domain. For text classification, we selected Chemprot (Kringelum et al., 2016), SciCite (Cohan et al., 2019), and SciERC (Luan et al., 2018). For NER, we used BC5CDR (Li et al., 2016), JNLPBA (Collier & Kim, 2004), and NCBI-Disease (Doğan et al., 2014). For QA, we chose the TriviaQA (Joshi et al., 2017) and SQuAD (Rajpurkar et al., 2016) datasets.

## 5 Results

### 5.1 Standard Fine-tuning

The results for standard fine-tuning of either full or reduced models are shown in Table 1. For the text classification and NER tasks, the reduced BERT-sized and larger models perform similarly to their fine-tuned counterparts on average, and substantially outperform the distilled models. The reduced distilled

---

[1]We note that for the encoder-decoder model T5, we consider caching only the middle layer of the *encoder*, which means that the speedups for this model will be smaller than (approximately half of) that of the other models we evaluate. We also consider 25% and 75% reduced models in Appendix A.

| Task | RoBERTa Large | | (Sci)BERT | | DeBERTa V2 XL | | T5 Large | | MiniLM L6-H768 | | MiniLM L6-H384 | | DistilBERT | |
|---|---|---|---|---|---|---|---|---|---|---|---|---|---|---|
| | Rdc | Full | Rdc | Full | Rdc | Full | Rdc | Full | Rdc | Full | Rdc | Full | Rdc | Full |
| ChemProt | 84.3 | 83.9 | 84.0 | 84.0 | 86.8 | 86.7 | 84.6 | 84.1 | 78.3 | 79.3 | 76.9 | 74.6 | 80.3 | 79.1 |
| SciCite | 85.0 | 85.5 | 86.6 | 86.0 | 85.2 | 84.4 | 86.3 | 84.9 | 84.5 | 84.6 | 83.7 | 82.8 | 84.1 | 84.0 |
| SciERC-Rel | 80.2 | 80.4 | 76.7 | 79.8 | 79.9 | 80.2 | 77.4 | 80.2 | 74.8 | 78.2 | 72.1 | 68.9 | 74.9 | 72.9 |
| Classification Avg. | 83.2 | **83.3** | 82.4 | **83.3** | **84.0** | 83.8 | 82.8 | **83.1** | 79.2 | **80.7** | **77.6** | 75.4 | **79.8** | 78.7 |
| bc5cdr | 90.0 | 90.4 | 90.7 | 91.3 | 91.3 | 91.8 | 90.7 | 89.9 | 87.8 | 87.5 | 85.9 | 88.3 | 88.3 | 88.7 |
| JNLPBA | 79.4 | 78.7 | 78.8 | 79.0 | 78.5 | 78.2 | 79.6 | 80.0 | 77.3 | 76.9 | 74.0 | 77.2 | 78.6 | 78.5 |
| NCBI-disease | 93.0 | 93.2 | 93.4 | 92.9 | 93.3 | 93.4 | 92.8 | 93.5 | 91.1 | 92.1 | 89.9 | 91.7 | 90.5 | 91.3 |
| NER Avg. | **87.5** | 87.4 | **87.7** | 87.7 | 87.7 | **87.8** | 87.7 | **87.8** | 85.4 | **85.5** | 83.3 | **85.7** | 85.8 | **86.2** |
| TriviaQA | 78.2 | 79.8 | 67.4 | 69.1 | 80.6 | 81.8 | 77.4 | 78.2 | 72.2 | 73.8 | 69.2 | 71.0 | 64.7 | 66.8 |
| SQuAD | 91.8 | 93.6 | 87.5 | 88.5 | 94.5 | 94.6 | 93.7 | 93.9 | 85.0 | 87.0 | 89.0 | 89.6 | 84.8 | 85.4 |
| QA Avg. | 85.0 | **86.7** | 77.5 | **78.8** | 87.5 | **88.2** | 85.5 | **85.9** | 78.6 | **80.4** | 79.1 | **80.3** | 74.8 | **76.1** |

Table 1: Test scores of reduced (Rdc) models on the text classification, NER, and QA tasks. **Bold** indicates the best average score between the reduced and fully fine-tuned (Full) versions of each model. Each score represents the average macro F-1 score of 10 runs for RoBERTa, BERT, and the distilled models. The F-1 score averages for DeBERTa and T5 were gathered from 5 runs. The standard errors for each score are shown in corresponding tables in Appendix A. For the ChemProt dataset, we report the micro F-1 scores instead, following past work (Beltagy et al., 2019). The reduced BERT-sized models generally offer similar performance to their full counterparts (scoring within 0.2% when averaged across RoBERTa and SciBERT for the six tasks), and substantially outperform the distilled models. For the QA datasets, we use BERT while we use SciBERT for the text classification and NER datasets since it outperforms BERT on these datasets (Beltagy et al., 2019). The reduced models yield a small accuracy drop for QA tasks.

models also perform well on those tasks compared to the distilled originals, on average, although there is more variance across models and tasks compared to BERT-sized models. We validate our fully fine-tuned baselines by comparing our results with prior work (Beltagy et al., 2019), finding that our scores land within 1.33% on average and typically score above the previous baselines.

For QA tasks, we found that fully fine-tuning works somewhat better than reduced configurations across all the explored models (Table 1). Generally, reduced configurations typically lag by 1 to 2 points in F-1 score. One possible hypothesis is that the QA datasets are generally much larger than the datasets we used for other tasks (100k-150k examples vs 4k-20k examples for text classification and NER); however, in additional experiments we found that subsampling the QA training sets to 5% of their original size only increased the gap, suggesting that dataset size does not explain the failure of reduced models on this task. We also validate our fully fine-tuned baselines for QA tasks by comparing our results with prior work (Yasunaga et al., 2022), finding that our scores land within 0.42% on average.

Additionally, we explored using lightweight multi-layer perceptrons (MLPs) as classifier heads, given their success in previous work. While Du et al. (2020) paired multi-task encoders with 2-layer MLPs, we paired frozen pretrained transformer models with 2-layer MLPs and found that they underperformed trainable layers dramatically, by 26% on average across the text classification and NER tasks.

## 5.2 Adapters

Our results for reduced adapter models are shown in Table 2. We see that in general, for all the models except for T5-Large, the adapter-based approaches are superior to standard fine-tuning on our tasks. Further, layer recycling remains effective with adapters. Compared to the full adapter baseline, the reduced adapters for RoBERTa-Large, BERT, and SciBERT models only show a 0.15-0.17% reduction in accuracy. Additionally, compared to the fully fine-tuned baseline, these reduced adapters models have a 0.19-0.23% reduction in

| Task | RoBERTa Large | | | (Sci)BERT | | | DeBERTa V2 XL | | | T5 Large | | |
|---|---|---|---|---|---|---|---|---|---|---|---|---|
| | Rdc + Half Adpt | Full Adpt | Full | Rdc + Half Adpt | Full Adpt | Full | Rdc + Half Adpt | Full Adpt | Full | Rdc + Half Adpt | Full Adpt | Full |
| ChemProt | 84.1 | 85.2 | 83.9 | 84.2 | 84.9 | 84.0 | 87.2 | 86.5 | 86.7 | 84.3 | 84.9 | 84.1 |
| SciCite | 82.4 | 82.9 | 85.5 | 85.5 | 84.6 | 86.0 | 84.6 | 85.0 | 84.4 | 85.3 | 84.5 | 84.9 |
| SciERC-Rel | 85.7 | 85.9 | 80.4 | 86.0 | 85.5 | 79.8 | 82.9 | 82.1 | 80.2 | 76.2 | 75.6 | 80.2 |
| Classification Avg. | 84.1 | **84.7** | 83.3 | **85.2** | 85.0 | 83.3 | **84.9** | 84.6 | 83.8 | 81.9 | 81.7 | **83.1** |
| bc5cdr | 90.0 | 90.6 | 90.4 | 90.0 | 90.9 | 91.3 | 90.7 | 91.1 | 91.8 | 79.9 | 85.7 | 89.9 |
| JNLPBA | 79.1 | 79.2 | 78.7 | 79.8 | 78.3 | 79.0 | 79.3 | 79.0 | 78.2 | 78.8 | 79.5 | 80.0 |
| NCBI-disease | 92.8 | 93.1 | 93.2 | 93.1 | 93.0 | 92.9 | 93.3 | 93.5 | 93.4 | 92.1 | 92.5 | 93.5 |
| NER Avg. | 87.3 | **87.6** | 87.4 | 87.6 | 87.4 | **87.7** | 87.8 | **87.9** | 87.8 | 83.6 | 85.9 | **87.8** |
| TriviaQA | 78.5 | 79.8 | 79.8 | 67.4 | 68.9 | 69.1 | 81.6 | 82.3 | 81.8 | 77.0 | 77.5 | 78.2 |
| SQuAD | 93.5 | 93.4 | 93.6 | 87.9 | 87.9 | 88.5 | 94.7 | 93.9 | 94.6 | 90.6 | 91.0 | 93.9 |
| QA Avg. | 86.0 | 86.6 | **86.7** | 77.6 | 78.4 | **78.8** | 88.1 | 88.1 | **88.2** | 83.8 | 84.3 | **85.9** |

Table 2: Test scores of reduced adapter (Rdc + Half Adpt) models on the text classification, NER, and QA tasks. **Bold** indicates the best average score between the reduced adapter, full adapter (Full Adpt), and fully fine-tuned (Full) versions of each model. Each score represents the average macro F-1 score of 10 runs for RoBERTa, BERT, and the distilled models. The F-1 score averages for DeBERTa and T5 were gathered from 5 runs. The standard errors for each score are shown in corresponding tables in Appendix A. For the ChemProt dataset, we report the micro F-1 scores instead, following past work (Beltagy et al., 2019). For the QA datasets, we use BERT while we use SciBERT for the text classification and NER datasets since it outperforms BERT on these datasets (Beltagy et al., 2019).

accuracy. Likewise, in contrast to the full fine-tuning results above, QA accuracy for the top-performing DeBERTa adapter model remains unchanged on average after layer recycling, with the reduced adapter model performing better on one QA task and worse on the other.[2]

### 5.3 Efficiency Analysis

To estimate the real-world benefit of recycling embeddings for different tasks, we provide a minimal PyTorch implementation of embedding recycling.[3] This implementation and the following results correspond to both the standard layer recycling approach and the adapter-based layer recycling approach since they follow parallel processes for gradient descent during training and computations during inference, despite the additional 6-8% of parameters added by the trainable adapters. To show that training times do not differ substantially, we also measured the training time the transformer models take to converge to their optimal weights. We found both approaches take approximately the same training time to complete training (Table 15).

We evaluated the impact of recycling embeddings on four different architectures and two different hardware platforms. For models, we considered two efficient transformer models (Two 6-layer MiniLMv2 (Wang et al., 2020; 2021) models with embeddings of size 384 and 768), a base model (BERT$_{\text{BASE}}$ uncased, 12 layers and 768 dimensional embeddings), and a large model (BERT$_{\text{LARGE}}$ uncased) with 24 layers and embeddings of size 1024. Intuitively, we expect larger models with more layers to benefit more from embedding recycling. For platforms, we ran our proof-of-concept implementation on an AWS cloud instance[4] equipped with an NVIDIA A10G accelerator, and on a NVIDIA A6000 within an on-premise server[5]. The former contains fewer

---

[2]We omit experiments with distilled models, as we found adapters to be ineffective on those models even without embedding recycling, scoring 19.4% worse on average than full fine-tuning for text classification and NER.

[3]Code and documentation available in supplementary material.

[4]`g5.2xlarge` instance with 8 cores and 32 GB of RAM.

[5]Intel-based system with 128 cores and 512 GB of RAM.

| Model | Baseline (NR) | Inference Time (ms/batch) Recycling | | Speedup | | Avg. F1 Loss using Recycling |
|---|---|---|---|---|---|---|
| | | FP32 Cache | FP16 Cache | NR vs FP32 | NR vs FP16 | |
| NVIDIA A10G | | | | | | |
| MiniLM$_{384}$ | $183 \pm 1$ | $154 \pm 22$ | $123 \pm 2$ | +21% | +67% | -0.2 |
| MiniLM$_{768}$ | $325 \pm 1$ | $201 \pm 11$ | $195 \pm 4$ | +56% | +66% | -0.4 |
| BERT$_{\text{BASE}}$ | $647 \pm 1$ | $351 \pm 1$ | $343 \pm 5$ | +84% | +88% | -0.3 |
| BERT$_{\text{LARGE}}$ | $1943 \pm 1$ | $1066 \pm 12$ | $1004 \pm 4$ | +86% | +93% | -0.2 |
| DeBERTa$_{\text{XLARGE}}$ | $1914 \pm 2$ | $1010 \pm 10$ | $985 \pm 8$ | +89% | +94% | -0.1 |
| NVIDIA A6000 | | | | | | |
| MiniLM$_{384}$ | $123 \pm 1$ | $105 \pm 5$ | $100 \pm 2$ | +18% | +23% | -0.2 |
| MiniLM$_{768}$ | $208 \pm 1$ | $161 \pm 8$ | $150 \pm 4$ | +29% | +38% | -0.4 |
| BERT$_{\text{BASE}}$ | $416 \pm 1$ | $269 \pm 6$ | $245 \pm 2$ | +55% | +59% | -0.3 |
| BERT$_{\text{LARGE}}$ | $1235 \pm 1$ | $662 \pm 10$ | $643 \pm 10$ | +86% | +92% | -0.2 |
| DeBERTa$_{\text{XLARGE}}$ | $1430 \pm 2$ | $777 \pm 6$ | $758 \pm 4$ | +84% | +89% | -0.1 |

Table 3: Average **inference** runtime comparison (in ms per batch, $\pm$ stdev over 7 runs) between vanilla encoders and models that cache embeddings on disk. We assume the cache is precomputed (see subsection 5.3). For all runs, cache the middle layer of the encoder; thus, maximum speedup is 100%. Overall, the larger the model, the higher the speedup from re-using representations. Further, accelerators with fewer execution units (A10G) benefit more from recycling embeddings. Finally, using half precision for embedding caching improves speed up across the board, while halving storage size. We also found that half-precision has a negligible effect on F1-scores at inference if you originally train the models using full-precision. In the rightmost column, we included the average F1 loss from using embedding recycling across our tasks.

execution units (72 vs 84), fewer tensor cores (288 vs 336), slower memory (600 vs 768 GB/s), and slower boost clock (1800 MHz vs 1695 MHz). However, it is much more efficient, being rated at 150W (compare with A6000's 300W power target). Therefore, the NVIDIA A10G accelerator presents a more realistic platform for embedding recycling, since it is more suitable for cost-efficient large-scale model deployments. Both machines are equipped with PCIe NVMe drives, which we use to cache embeddings to recycle.

Table 3 shows the results of caching embeddings to recycle on disk. Because inference time varies across tasks depending on dataset properties, such as length of sequences and number of samples, we control our experiments by simulating a sequence classification task on QASPER (Dasigi et al., 2021), which includes the full-text of over a thousand academic manuscripts.[6] Further, we run all models with a fixed batch size of 128 and maximum sequence length of 512; for all models, we reduce exactly half of their layers by recycling, which results in a maximum theoretical speed-up of 100%. A run over the corpus consists of 335 batches, and we average results over seven runs. Overall, we found that in practice all models benefit from embedding recycling, achieving an average speedup ranging from 18 to 86%. Unsurprisingly, larger models benefit more from recycling than smaller ones; this is due to the fact that loading embeddings cached on disk adds a small latency penalty to a model run, which is more noticeable in the case of smaller models. For example, we achieve an 84% speedup when running BERT$_{\text{BASE}}$ with embedding recycling on an A10G GPU, which is roughly equivalent to the latency of a MiniLM$_{768}$ model without recycling (351 vs 325 ms per batch on average); this result would us allow to run more accurate models while maintaining the efficiency of shallower architectures.

We note that in our inference-time speedup measurements, we assume that the cache is already precomputed. This corresponds to our target setting in which new tasks and models are executed over the same text that has been processed previously. Because the cost to write the cache to disk is approximately equal to a single inference pass over the corpus, if we perform $t$ total inference passes for different models/tasks given the same precomputed cache, the total amortized speedup will be approximately $\frac{t}{t+1}$ of the values we report in

---

[6]Because the bulk of computation for a transformer model is done in its encoder and not in the task-specific heads, inference time is similar regardless of whether the model is used for sequence classification, tagging, or question answering.

| Model | Training (ms/batch, amortized over **6 epochs**) | | | | Speedup | | |
|---|---|---|---|---|---|---|---|
| | **No Recycling** (NR) | **Model Frozen** (F) | **Saving** + **Recycling** (SR) | **Only Recycling** (R) | **NR** vs **SR** | **F** vs **SR** | **NR** vs **R** |
| | | | NVIDIA A10G | | | | |
| MiniLM$_{384}$ | $51 \pm 1$ | $30 \pm 1$ | $32 \pm 6$ | $25 \pm 4$ | +59% | -7% | +104% |
| MiniLM$_{768}$ | $90 \pm 4$ | $56 \pm 1$ | $50 \pm 4$ | $45 \pm 3$ | +80% | +12% | +100% |
| BERT$_{BASE}$ | $173 \pm 2$ | $112 \pm 1$ | $90 \pm 4$ | $87 \pm 3$ | +92% | +24% | +99% |
| BERT$_{LARGE}$ | $347 \pm 1$ | $246 \pm 1$ | $181 \pm 2$ | $176 \pm 2$ | +92% | +36% | +97% |
| DeBERTa$_{XLARGE}$ | $380 \pm 2$ | $286 \pm 1$ | $199 \pm 1$ | $194 \pm 1$ | +91% | +44% | +96% |
| | | | NVIDIA A6000 | | | | |
| MiniLM$_{384}$ | $41 \pm 1$ | $24 \pm 1$ | $26 \pm 5$ | $22 \pm 3$ | +55% | -8% | +81% |
| MiniLM$_{768}$ | $61 \pm 1$ | $38 \pm 1$ | $40 \pm 5$ | $34 \pm 3$ | +52% | -5% | +82% |
| BERT$_{BASE}$ | $117 \pm 1$ | $78 \pm 1$ | $60 \pm 3$ | $58 \pm 2$ | +94% | +30% | +102% |
| BERT$_{LARGE}$ | $326 \pm 2$ | $212 \pm 1$ | $167 \pm 2$ | $161 \pm 1$ | +96% | +26% | +103% |
| DeBERTa$_{XLARGE}$ | $359 \pm 2$ | $250 \pm 1$ | $184 \pm 1$ | $178 \pm 1$ | +95% | +35% | +102% |

Table 4: Average **training** runtime comparison (in ms per batch, $\pm$ stdev over 7 runs) between vanilla encoders and models that cache embeddings on disk. For all runs, we cache the middle layer of the encoder; thus, theoretical speedup is 100%. Time per batch is amortized over 6 epochs ($2,000$ steps), the lowest number to convergence over all datasets (c.r.f. Table 15). We present results in four settings: no recycling (NR), freezing ½ of the layers during training (F), 1 training epoch during which embeddings are saved to disk followed by 5 epochs where recycling is enabled (SR), and 6 epochs where embeddings are already saved (R). Overall, we found that embedding recycling speeds up training even when embeddings need to be cached to disk during the first pass. Compared to freezing, saving and recycling improves training time for all but MiniLM models (F vs SR).

Table 3. Thus, as the number of inference passes to be run increases, the total amortized speedup including the cost to write the cache will approach the values reported in the table.

Table 3 also includes results when storing embeddings using half precision (that is, cache embeddings in FP16 rather FP32). When using half precision, we observed improvements for all models and hardware, ranging from +8% to +46%. Storing cached embeddings in FP16 has virtually no impact on performance, as it changes predicted scores by typically $10^{-3} - 10^{-5}$ across all tasks evaluated in this work.

We also note that less capable hardware benefits more from caching embeddings. For example, BERT$_{BASE}$ achieves a speedup of 84% on an A10G GPU, while on A6000, the speedup is a more modest 55%. This is an expected result: fewer and slower execution cores/accelerator memory impact overall model latency. Further, we note that, despite the smaller relative gains, the more powerful GPU is always faster in absolute terms compared with the less capable one.

It is important to note that these gaps from maximum achievable speedup are only observed when performing *inference*; for *training*, we observe almost perfect speed-up for all models and hardware configurations barring MiniLM models on the machine equipped with a A6000 GPU ("NR vs R" column in Table 4). For example, BERT$_{BASE}$ requires $17.38 \pm 1.32$ ms/batch[7] without recycling, compared to $8.67 \pm 2.18$ ms/batch when recycling. Even when considering the additional time to cache embeddings to disk during the first pass, embedding recycling still achieves close to optimum speedup on all models except MiniLMs, where its gains hover between 52% and 82% ("NR vs SR" column in Table 4). When training for just 6 epochs (or roughly $2,000$ steps), recycling embeddings is faster than simply freezing half of the parameters for all models but MiniLM ("F vs SR" column in Table 4); this is due to the relatively higher cost of caching layers to disk in case of smaller models. In these cases, we empirically found that recycling achieves faster training time than freezing after 12 epochs or $4,000$ training steps; since smaller models typically require more epochs to converge, we conclude that recycling is generally preferable to partially freezing a model during training.

---

[7] When training, we use a batch size of 16

We also benchmarked the storage requirements of recycling embeddings. For a sequence of 512 tokens and a hidden model dimension of 768, caching embeddings requires 1.6 MB. In practical terms, this translates to 15.5 MB per paper in QASPER (papers are, on average 4,884 WordPiece tokens in length). Besides the storage needs, NVMe disks, while fast, introduce additional latency compared to RAM. For example, $BERT_{BASE}$ achieves an average latency of $351 \pm 1$ ms/batch when caching on disk (84% speedup), compared to just $334 \pm 1$ ms/batch when using memory (94% speedup). To reduce the impact of this latency penalty, our implementation supports *pre-fetching* of future embeddings: when processing a sequence of inputs, such as sentences in a manuscript, it loads embeddings for tokens ahead of the sequence inference is currently being run on. This optimization reduces the time accelerators wait for data to be available for inference; for example, in the case of $BERT_{BASE}$ on A10G, disabling pre-fetching raised inference inference time to $374 \pm 1$ ms/batch (vs $351 \pm 1$ ms/batch with pre-fetching). Therefore in this section, all results are reported with prefetching enabled.

## 6 Discussion

### 6.1 Cross-model Embedding Reuse

| | | RoBERTa-Large + MiniLM L6-H768 | MiniLM L6-H768 | BERT + DistilBERT | DistilBERT |
|---|---|---|---|---|---|
| **Chemprot** | Micro F-1 | 78.9 (0.3) | 79.3 (0.3) | 77.8 (0.4) | 79.1 (0.5) |
| | Macro F-1 | 52.2 (0.2) | 52.6 (0.4) | 51.2 (0.5) | 52.6 (0.3) |
| **SciCite** | Micro F-1 | 85.2 (0.3) | 86.0 (0.2) | 85.7 (0.1) | 85.5 (0.1) |
| | Macro F-1 | 83.8 (0.3) | 84.6 (0.2) | 84.2 (0.1) | 84.0 (0.1) |
| **SciERC-Rel** | Micro F-1 | 85.1 (0.4) | 86.3 (0.2) | 83.8 (0.2) | 83.5 (0.4) |
| | Macro F-1 | 76.2 (0.8) | 78.2 (0.6) | 73.6 (0.6) | 72.9 (0.7) |
| **Text Classification Average Score** | | 76.9 | **77.8** | 76.0 | **76.3** |

Table 5: Cross-Model Recycling Results for RoBERTa+MiniLM-L6H768 and BERT+DistilBERT configurations. **Bold** indicates the best average score between the cross-model recycling and fully finetuned versions of each model. Each score represents the average score of 10 runs, with the standard errors for each score in parentheses.

The experiments in section 5 focus on caching activations from a pretrained model and then re-using those for fine-tuning and inference with the same model. An alternative potential use-case of ER involves caching activations from a more expensive, larger model once and re-using those downstream within a cheaper model multiple times. Here, the goal is not to improve efficiency of the downstream model, but instead to improve its accuracy by introducing the helpful contextual embedding signal from the larger model. However, as we will show, a straightforward implementation of this strategy did not offer improvements in our experiments.

We experiment with reusing precomputed embeddings from one source model $F$ in a consumer model $F'$ that has a different transformer architecture but the same tokenization vocabulary. During the caching step for the source model, the activations of the *final* transformer layer $h_{c \in \mathcal{C}}^N$ are stored for each input $c$ of the selected corpus $C$. During the fine-tuning phase of the consumer model $F'$, these stored activations are transformed through a learned 2-layer MLP and added to the input embeddings of $F'$.

Using our text classification tasks, we tried two frameworks for pairing large language model embeddings with compact models. In our first framework, we use a frozen RoBERTa model as our source model and a MiniLM-6L-H768 model as our consumer model. In our second framework, we use a frozen BERT-base model as our source model and a DistilBERT model as our consumer model. We use a ReLU activation in the 2-layer MLP. In our testing, we also explored using a single linear layer rather than an MLP but found an MLP achieved better performance on the development sets for the text classification tasks. The results of our experiments are shown in Table 5. Overall, the larger model's contextual representations do not improve the smaller model's accuracy; in fact adding them decreases the average F1 score by 0.3-0.9 points.

### 6.2 GLUE Results

| | DeBERTa V2 XL | | | |
|---|---|---|---|---|
| | Rdc + Half Adpt | Full Adpt | Rdc | Full |
| CoLA | 70.9 | 71.3 | 70.8 | 71.2 |
| SST-2 | 96.9 | 97.1 | 97.1 | 97.4 |
| Single Sentence Avg. | 83.9 | **84.2** | 84.0 | **84.3** |
| MRPC | 93.9 | 94.0 | 93.4 | 93.9 |
| STS-B | 92.4 | 92.7 | 92.5 | 92.8 |
| Similarity and Paraphrase Avg. | 93.2 | **93.4** | 93.0 | **93.4** |
| MNLI | 91.6 | 92.0 | 91.0 | 91.4 |
| QNLI | 95.0 | 95.1 | 94.1 | 94.8 |
| NLI Avg. | 93.3 | **93.6** | 92.6 | **93.1** |

Table 6: Test scores of reduced (Rdc) and reduced adapter (Rdc + Half Adpt) models on the GLUE tasks for DeBERTa V2 XL. **Bold** indicates the best average score between the reduced and fully fine-tuned (Full) versions for both the standard and adapter-based configurations. Each score represents the average macro F-1 score of 5 runs. Our results for MNLI correspond to MNLI-matched. For MRPC, MNLI, and QNLI, we report the macro F1 scores. For CoLA, we report the Matthews correlation coefficient (MCC). For SST-2, we report the accuracy. For STS-B, we report the Pearson's correlation coefficient.

For our best-performing model DeBERTa v2 XL, we also provide further experiments on datasets from the GLUE benchmark (Wang et al., 2018), to allow easier comparison against speedup techniques from previous work. In this revision we present preliminary results on the CoLA, SST-2, MRPC, STS-B, MNLI, and QNLI tasks from GLUE. For our experiments, we tried both our standard reduced models and our reduced adapter models. We found that embedding recycling was successful across the GLUE tasks, with a small loss in F1 scores in return for a significant increase in both training and inference time as outlined in Table 3 and Table 4. We note that due to the high computational cost of these experiments, we take existing hyperparameter settings from previous work that worked well for the full models, and also use these for reduced models. Further hyperparameter optimization of the reduced models might improve performance.

### 6.3 Directions for Future Work

Our experiments raise several questions that could be answered in future studies, and embedding recycling contains a much larger space of potential techniques than those we investigate here. Future work could proceed along various lines, including:

- Our layer recycling strategy is a straightforward ER approach, but previous work has suggested that weighted pooling across layers can perform better compared to any single layer in many cases (Liu et al., 2019a; Du et al., 2020). Recycling pooled activations may offer improved results. What is the best way to capture and store the syntactic and semantic knowledge encoded in the activations of a model for later recycling?

- Our experiments show that the right recycling approach may be task-specific and model-specific. For example, with standard fine-tuning as shown in Table 7, caching layer 12 in RoBERTa-large is most effective for NER and text classification, whereas it is not effective for QA (but layer 6 performs much better). Which embeddings to retrieve and recycle for a task, and the right architecture (e.g. number of layers) to use when consuming the recycled embeddings, represents a large decision space. Methods that can help practitioners automatically choose among public or private shared embedding

sets and associated model designs, given their task and objectives for accuracy and computational cost, may be important to make ER an effective practical tool.

- We present results with encoder-only and encoder-decoder models, on classification tasks. Determining the approach is effective for generative tasks and autoregressive models is an important question for future work.

- Other than recycling, a variety of other inference-speedup techniques exist for large neural models. While we show that ER can be effective when coupled with distillation, whether other techniques like quantization and early exiting remain effective in combination with ER is an open question.

- We focus on the setting where the exact same text, at the length of a full document, is being reused for multiple tasks. In practice, we may often perform a task on text that is *similar* to but not exactly the same as one for which we have cached embeddings (e.g., a Wikipedia page that has been revised). Further, even a completely new document will have similarities and overlapped spans with previously processed ones. Studying ER in these settings, e.g. through a combination of layer recycling and the SkipBERT approach which can apply to unseen passages via cached n-grams (Wang et al., 2022), is an area of future work.

- Finally, our techniques for cross-model embedding reuse were not effective in our experiments. However, using rich contextualized embeddings from a large model to help power many smaller downstream task models is an important setting for ER, since it provides a powerful way to amortize the expense of running a large model. Developing and evaluating new approaches for this setting is an important item for future work.

## 7 Conclusion

We have presented embedding recycling, a general technique for reusing previous activations of neural language models to improve the efficiency of future training and inference. We show how a simple technique of caching a layer of activations in a pretrained model is effective. We validate our approach in experiments across eight tasks and seven model architectures. We find that recycling typically has small or no impacts to accuracy on average, but does yield substantial throughput increases demonstrated through a careful efficiency analysis. We also discuss several open challenges for future work.

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

# A   Experimental Setup and Additional Results

## A.1   Fine-tuning Transformer Models

The candidate transformer models are fine-tuned using configurations suggested by Devlin et al. (2019), Ding et al. (2022) and Houlsby et al. (2019). For text classification, we feed the final hidden state of the [CLS] token into a linear classification layer. For NER and QA, we feed the final hidden states of each token into a linear classification layer with a softmax output.

For all of the models, we apply a dropout of 0.1 to the transformer outputs and optimize for cross entropy loss using Adam (Kingma & Ba, 2015). We employ a batch size of 32 across all tasks. We fine-tune using early stopping with a patience of 10, using a validation set for calculating loss for each epoch. We use a linear warmup followed by linear decay for training (Howard & Ruder, 2018), testing the following learning rate options: 1e-3, 2e-3, 1e-4, 2e-4, 1e-5, 2e-5, 5e-5, and 5e-6. For the text classification and NER datasets, we select the best performing learning rate for each transformer model on the development set and report the corresponding test results. For the QA datasets, we select the best performing learning rate for each transformer model on the training set and report the corresponding results on the validation set. Additionally, for the adapter modules used in certain model configurations, we test bottleneck dimensions as part of our hyperparameter search: 24, 64, and 256.

## A.2   Adapter-based Models

Here, we used frozen RoBERTa-Large (Liu et al., 2019b), SciBERT (Beltagy et al., 2019), and BERT models but added adapter modules (Houlsby et al., 2019) only on the latter half of the transformer layers. Only the adapters and the linear classifier attached to the model output were fine-tuned for the text classification, NER, and QA tasks.

We found that the best hyperparameter configuration was generally a bottleneck dimension of 256 and a learning rate of either 1e-4 or 2e-4.

## A.3   Software and Hardware

For implementation, we use the v4.19 version of the Transformers library (Wolf et al., 2019), the v0.4 version of the OpenDelta library (Ding et al., 2022), and the v1.11 version of the Pytorch library (Paszke et al., 2019). We conduct our experiments using NVIDIA RTX A6000 GPUs and NVIDIA A10G GPUs with CUDA v11.5.

| | | RoBERTa-Large | | | | | |
|---|---|---|---|---|---|---|---|
| | | Reduced + Half Adpt | Full Adapters | 6 Layers Reduced | 12 Layers Reduced | 18 Layers Reduced | Fully Finetuned |
| **ChemProt** | Micro F-1 | 84.1 (0.4) | **85.2 (0.3)** | 84.2 (0.3) | 84.3 (0.2) | 82.0 (0.2) | 83.9 (0.3) |
| | Macro F-1 | **60.8 (0.7)** | 57.5 (0.7) | 56.4 (0.4) | 56.5 (0.3) | 54.5 (0.5) | 56.5 (0.4) |
| **SciCite** | Micro F-1 | 85.2 (0.3) | 85.6 (0.5) | 86.2 (0.2) | 86.2 (0.2) | 86.2 (0.2) | **86.8 (0.2)** |
| | Macro F-1 | 82.4 (0.4) | 82.9 (0.6) | 84.9 (0.2) | 85.0 (0.2) | 85.0 (0.2) | **85.5 (0.2)** |
| **SciERC-Rel** | Micro F-1 | 89.0 (0.5) | **89.3 (0.6)** | 87.1 (0.4) | 86.8 (0.4) | 86.1 (0.2) | 87.3 (0.4) |
| | Macro F-1 | 85.7 (0.7) | **85.9 (0.9)** | 79.4 (0.7) | 80.2 (0.8) | 76.2 (0.4) | 80.4 (0.6) |
| **Text Classification Average Score** | | **81.2** | 81.1 | 79.7 | 79.8 | 78.3 | 80.1 |
| **bc5cdr** | Micro F-1 | 97.4 (0.0) | **97.6 (0.0)** | 97.2 (0.3) | 97.4 (0.0) | 97.3 (0.0) | 97.5 (0.0) |
| | Macro F-1 | 90.0 (0.0) | **90.6 (0.0)** | 89.0 (1.2) | 90.0 (0.0) | 89.5 (0.1) | 90.4 (0.1) |
| **JNLPBA** | Micro F-1 | 93.8 (0.0) | 93.8 (0.0) | 93.8 (0.0) | **93.9 (0.0)** | 93.7 (0.0) | 93.7 (0.1) |
| | Macro F-1 | 79.1 (0.1) | 79.2 (0.2) | 79.3 (0.1) | **79.4 (0.1)** | 79.0 (0.1) | 78.7 (0.3) |
| **NCBI-disease** | Micro F-1 | 98.5 (0.0) | **98.6 (0.0)** | 98.5 (0.0) | 98.5 (0.0) | 98.4 (0.0) | **98.6 (0.0)** |
| | Macro F-1 | 92.8 (0.1) | 93.1 (0.1) | 93.0 (0.1) | 93.0 (0.1) | 92.4 (0.1) | **93.2 (0.1)** |
| **NER Average Score** | | 91.9 | **92.1** | 91.8 | 92.0 | 91.7 | 92.0 |
| **TriviaQA** | Micro F-1 | 75.3 (0.1) | **76.8 (0.2)** | 76.6 (0.2) | 75.1 (0.1) | 70.8 (0.1) | 76.7 (0.1) |
| | Macro F-1 | 78.5 (0.1) | **79.8 (0.1)** | 79.7 (0.2) | 78.2 (0.1) | 73.8 (0.1) | **79.8 (0.1)** |
| **SQuAD** | Micro F-1 | 87.0 (0.1) | 86.7 (0.0) | 86.2 (0.0) | 84.7 (0.0) | 79.3 (0.0) | **87.4 (0.0)** |
| | Macro F-1 | 93.5 (0.1) | 93.4 (0.0) | 92.8 (0.0) | 91.8 (0.0) | 87.8 (0.0) | **93.6 (0.0)** |
| **QA Average Score** | | 83.6 | 84.1 | 83.8 | 82.4 | 77.9 | **84.3** |

Table 7: RoBERTa Results for Reduced Models. **Bold** indicates the best average score between the standard reduced, adapter-based reduced, and fully fine-tuned versions of each model. **Reduced + Half Adpt** indicates adapters on the transformer layers of a fully frozen reduced model, where the earlier half of transformer layers were removed and their activations cached. **Full Adapters** indicates adapters on all transformer layers of a fully frozen model. Each score represents the average score of 10 runs, with the standard errors for each score in parentheses.

| | | SciBERT | | | | | |
|---|---|---|---|---|---|---|---|
| | | Reduced + Half Adpt | Full Adapters | 3 Layers Reduced | 6 Layers Reduced | 9 Layers Reduced | Fully Finetuned |
| **ChemProt** | Micro F-1 | 84.2 (0.3) | **84.9 (0.4)** | 83.8 (0.4) | 84.0 (0.2) | 81.9 (0.2) | 84.0 (0.3) |
| | Macro F-1 | 56.9 (0.8) | 54.8 (0.4) | 56.5 (0.5) | **57.0 (0.3)** | 54.3 (0.3) | 56.3 (0.4) |
| **SciCite** | Micro F-1 | 86.6 (0.2) | 85.8 (0.1) | 87.1 (0.1) | **87.6 (0.1)** | 87.4 (0.1) | 87.1 (0.2) |
| | Macro F-1 | 85.5 (0.3) | 84.6 (0.1) | 86.1 (0.1) | **86.6 (0.1)** | 86.2 (0.1) | 86.0 (0.2) |
| **SciERC-Rel** | Micro F-1 | **89.4 (0.4)** | 88.5 (0.6) | 86.6 (0.3) | 86.1 (0.2) | 85.4 (0.2) | 86.3 (0.2) |
| | Macro F-1 | **86.0 (0.7)** | 85.5 (0.6) | 77.6 (0.5) | 76.7 (0.3) | 76.2 (0.4) | 79.8 (0.5) |
| **Text Classification Average Performance** | | **81.4** | 80.7 | 79.6 | 79.7 | 78.6 | 79.9 |
| **bc5cdr** | Micro F-1 | 97.5 (0.0) | **97.7 (0.1)** | 97.7 (0.0) | 97.6 (0.0) | 97.5 (0.0) | **97.7 (0.0)** |
| | Macro F-1 | 90.0 (0.0) | 90.9 (0.1) | 91.0 (0.1) | 90.7 (0.0) | 90.2 (0.1) | **91.3 (0.0)** |
| **JNLPBA** | Micro F-1 | **94.0 (0.0)** | 93.5 (0.0) | 93.6 (0.1) | 93.7 (0.1) | 93.8 (0.0) | 93.6 (0.1) |
| | Macro F-1 | **79.8 (0.0)** | 78.3 (0.2) | 78.6 (0.4) | 78.8 (0.2) | 79.0 (0.1) | 79.0 (0.2) |
| **NCBI-disease** | Micro F-1 | **98.6 (0.0)** | 98.5 (0.0) | 98.5 (0.0) | **98.6 (0.0)** | 98.5 (0.0) | 98.5 (0.0) |
| | Macro F-1 | 93.1 (0.1) | 93.0 (0.1) | 92.9 (0.1) | **93.4 (0.1)** | 93.1 (0.1) | 92.9 (0.1) |
| **NER Average Performacne** | | **92.2** | 92.0 | 92 | 92.1 | 92 | **92.2** |

Table 8: SciBERT text classification and NER results for Reduced Models. **Bold** indicates the best average score between the standard reduced, adapter-based reduced, and fully fine-tuned versions of each model. **Reduced + Half Adpt** indicates adapters on the transformer layers of a fully frozen reduced model, where the earlier half of transformer layers were removed and their activations cached. **Full Adapters** indicates adapters on all transformer layers of a fully frozen model. Each score represents the average score of 10 runs, with the standard errors for each score in parentheses. QA tasks are not included since SciBERT was pretrained for scientific datasets.

| | | BERT | | | | | |
|---|---|---|---|---|---|---|---|
| | | Reduced + Half Adpt | Full Adapters | 3 Layers Reduced | 6 Layers Reduced | 9 Layers Reduced | Fully Finetuned |
| **TriviaQA** | Micro F-1 | 63.9 (0.5) | 65.5 (0.1) | 65.7 (0.1) | 64.1 (0.2) | 61.4 (0.1) | **66.0 (0.1)** |
| | Macro F-1 | 67.4 (0.5) | 68.9 (0.1) | 68.9 (0.1) | 67.4 (0.1) | 64.8 (0.1) | **69.1 (0.1)** |
| **SQuAD** | Micro F-1 | 80.2 (0.1) | 80.2 (0.0) | 80.8 (0.1) | 79.5 (0.1) | 75.4 (0.1) | **81.1 (0.1)** |
| | Macro F-1 | 87.9 (0.1) | 87.9 (0.0) | 88.4 (0.1) | 87.5 (0.1) | 84.8 (0.1) | **88.5 (0.0)** |
| **QA Average Scores** | | 74.9 | 75.6 | 76.0 | 74.6 | 71.6 | **76.2** |

Table 9: BERT QA Results for Reduced Models. **Bold** indicates the best average score between the standard reduced, adapter-based reduced, and fully fine-tuned versions of each model. **Reduced + Half Adpt** indicates adapters on the transformer layers of a fully frozen reduced model, where the earlier half of transformer layers were removed and their activations cached. **Full Adapters** indicates adapters on all transformer layers of a fully frozen model. Each score represents the average score of 10 runs, with the standard errors for each score in parentheses.

| | | DeBERTaV2 XL | | | | | |
|---|---|---|---|---|---|---|---|
| | | Reduced + Half Adpt | Full Adapters | 6 Layers Reduced | 12 Layers Reduced | 18 Layers Reduced | Fully Finetuned |
| **ChemProt** | Micro F-1 | **87.2 (0.1)** | 86.5 (0.2) | **87.2 (0.2)** | 86.8 (0.4) | 86.4 (0.2) | 86.7 (0.9) |
| | Macro F-1 | 56.7 (0.5) | 55.6 (0.6) | **59.6 (0.2)** | 59.5 (0.5) | 59.2 (0.3) | 59.0 (1.1) |
| **SciCite** | Micro F-1 | 85.8 (0.4) | **86.4 (0.4)** | 86.0 (0.1) | 86.3 (0.2) | 86.2 (0.3) | 85.9 (0.2) |
| | Macro F-1 | 84.6 (0.4) | 85.0 (0.5) | 84.6 (0.1) | **85.2 (0.1)** | 85.0 (0.3) | 84.4 (0.2) |
| **SciERC-Rel** | Micro F-1 | **88.6 (0.5)** | 88.0 (0.4) | 88.3 (0.2) | 87.5 (0.1) | 86.6 (0.3) | 88.0 (0.4) |
| | Macro F-1 | **82.9 (0.8)** | 82.1 (0.8) | 80.5 (0.5) | 79.9 (0.3) | 78.0 (0.4) | 80.2 (0.5) |
| **Text Classification Average Score** | | **81.0** | 80.6 | **81.0** | 80.9 | 80.2 | 80.7 |
| **bc5cdr** | Micro F-1 | 97.6 (0.0) | 97.7 (0.0) | 97.4 (0.3) | 97.7 (0.0) | 97.6 (0.0) | **97.9 (0.0)** |
| | Macro F-1 | 90.7 (0.1) | 91.1 (0.1) | 89.5 (1.4) | 91.3 (0.0) | 90.9 (0.0) | **91.8 (0.1)** |
| **JNLPBA** | Micro F-1 | 93.6 (0.0) | 93.4 (0.0) | **93.7 (0.1)** | 93.7 (0.0) | 93.6 (0.0) | **93.7 (0.0)** |
| | Macro F-1 | **79.3 (0.1)** | 79.0 (0.1) | 78.5 (0.3) | 78.5 (0.2) | 77.8 (0.1) | 78.2 (0.1) |
| **NCBI-disease** | Micro F-1 | 98.3 (0.0) | 98.4 (0.0) | **98.6 (0.0)** | **98.6 (0.0)** | 98.5 (0.0) | **98.6 (0.0)** |
| | Macro F-1 | 93.3 (0.1) | **93.5 (0.2)** | 93.1 (0.1) | 93.3 (0.1) | 92.8 (0.1) | 93.4 (0.1) |
| **NER Average Score** | | 92.1 | 92.2 | 91.8 | 92.2 | 91.9 | **92.3** |
| **TriviaQA** | Micro F-1 | 78.6 (0.2) | **79.1 (0.2)** | 77.9 (0.2) | 77.4 (0.2) | 77.0 (0.2) | 78.5 (0.1) |
| | Macro F-1 | 81.6 (0.1) | **82.3 (0.2)** | 81.2 (0.1) | 80.6 (0.1) | 80.1 (0.2) | 81.8 (0.1) |
| **SQuAD** | Micro F-1 | 88.6 (0.0) | 87.2 (0.1) | 88.6 (0.1) | **88.7 (0.0)** | 87.1 (0.0) | 88.5 (0.1) |
| | Macro F-1 | **94.7 (0.0)** | 93.9 (0.0) | 94.6 (0.0) | 94.5 (0.0) | 93.5 (0.0) | 94.6 (0.0) |
| **QA Average Score** | | **85.9** | 85.6 | 85.6 | 85.3 | 84.4 | 85.8 |

Table 10: DeBERTaV2-XL Results for Reduced Models. **Bold** indicates the best average score between the standard reduced, adapter-based reduced, and fully fine-tuned versions of each model. **Reduced + Half Adpt** indicates adapters on the transformer layers of a fully frozen reduced model, where the earlier half of transformer layers were removed and their activations cached. **Full Adapters** indicates adapters on all transformer layers of a fully frozen model. Each score represents the average score of 5 runs, with the standard errors for each score in parentheses.

| | | T5 Large | | | | | |
|---|---|---|---|---|---|---|---|
| | | Reduced + Half Adpt | Full Adapters | 6 Layers Frozen | 12 Layers Reduced | 18 Layers Reduced | Fully Finetuned |
| **ChemProt** | Micro F-1 | 84.3 (0.6) | 84.9 (0.6) | 84.7 (0.6) | 84.6 (0.6) | **85.0 (0.1)** | 84.1 (0.8) |
| | Macro F-1 | 57.2 (0.7) | **58.0 (0.8)** | 56.2 (0.7) | 56.2 (0.7) | 57.4 (0.1) | 56.1 (0.7) |
| **SciCite** | Micro F-1 | 86.7 (0.3) | 86.2 (0.3) | 87.4 (0.2) | 87.6 (0.1) | **88.0 (0.2)** | 86.4 (0.2) |
| | Macro F-1 | 85.3 (0.4) | 84.5 (0.4) | 86.0 (0.2) | 86.3 (0.2) | **86.9 (0.2)** | 84.9 (0.2) |
| **SciERC-Rel** | Micro F-1 | 85.6 (0.4) | 85.2 (0.1) | 84.3 (0.3) | 86.8 (0.4) | 83.4 (0.7) | **87.4 (0.5)** |
| | Macro F-1 | 76.2 (1.0) | 75.6 (0.2) | 73.6 (0.9) | 77.4 (0.7) | 72.2 (1.0) | **80.2 (1.1)** |
| **Text Classification Average Score** | | 79.2 | 79.1 | 78.7 | 79.8 | 78.8 | **79.9** |
| **bc5cdr** | Micro F-1 | 93.8 (0.6) | 95.7 (0.7) | **97.7 (0.7)** | 97.4 (0.3) | 95.4 (0.8) | 97.5 (0.2) |
| | Macro F-1 | 79.9 (1.0) | 85.7 (1.1) | **91.1 (0.5)** | 90.7 (1.1) | 89.3 (1.0) | 89.9 (0.8) |
| **JNLPBA** | Micro F-1 | 93.9 (0.4) | 93.8 (0.1) | 93.8 (0.0) | 94.0 (0.0) | 93.9 (0.0) | **94.2 (0.0)** |
| | Macro F-1 | 78.8 (0.6) | 79.5 (0.2) | 78.8 (0.1) | 79.6 (0.1) | 79.3 (0.0) | **80.0 (0.0)** |
| **NCBI-disease** | Micro F-1 | 97.8 (0.0) | 98.5 (0.0) | 98.5 (0.0) | 98.5 (0.0) | 98.4 (0.0) | **98.6 (0.0)** |
| | Macro F-1 | 92.1 (0.2) | 92.5 (0.2) | 93.1 (0.1) | 92.8 (0.0) | 92.2 (0.1) | **93.5 (0.0)** |
| **NER Average Score** | | 89.4 | 90.9 | 92.2 | 92.2 | 91.4 | **92.3** |
| **TriviaQA** | Micro F-1 | 68.2 (0.2) | **68.8 (0.2)** | 67.0 (0.0) | 66.9 (0.0) | 63.9 (0.0) | 68.7 (0.0) |
| | Macro F-1 | 77.0 (0.1) | 77.5 (0.1) | 77.5 (0.0) | 77.3 (0.0) | 74.8 (0.0) | **78.0 (0.0)** |
| **SQuAD** | Micro F-1 | 81.2 (0.1) | 82.0 (0.1) | 86.6 (0.1) | 86.3 (0.6) | 85.2 (0.4) | **86.7 (0.4)** |
| | Macro F-1 | 90.6 (0.1) | 91.0 (0.1) | 93.8 (0.0) | 93.7 (0.3) | 92.8 (0.2) | **93.9 (0.3)** |
| **QA Average Score** | | 79.2 | 79.8 | 81.2 | 81.0 | 79.2 | **81.8** |

Table 11: T5 Large Results for Reduced Models. **Bold** indicates the best average score between the standard reduced, adapter-based reduced, and fully fine-tuned versions of each model. **Reduced + Half Adpt** indicates adapters on the encoder and decoder transformer layers of a fully frozen reduced model, where the earlier half of the encoder layers were removed and their activations cached. **Full Adapters** indicates adapters on all encoder and decoder transformer layers of a fully frozen model. Each score represents the average score of 5 runs, with the standard errors for each score in parentheses.

| | | DistilBERT | | | |
|---|---|---|---|---|---|
| | | **2 Layers Reduced** | **3 Layers Reduced** | **4 Layers Reduced** | **Fully Fine-tuned** |
| **ChemProt** | Micro F-1 | 79.1 (0.4) | **80.3 (0.1)** | 79.0 (0.2) | 79.1 (0.5) |
| | Macro F-1 | 52.1 (0.5) | 51.6 (0.6) | 51.6 (0.4) | **52.6 (0.3)** |
| **SciCite** | Micro F-1 | 85.7 (0.1) | 85.6 (0.1) | **85.8 (0.1)** | 85.5 (0.1) |
| | Macro F-1 | **84.3 (0.1)** | 84.1 (0.1) | 84.2 (0.1) | 84.0 (0.1) |
| **SciERC-Rel** | Micro F-1 | 84.3 (0.3) | 84.5 (0.3) | **84.6 (0.2)** | 83.5 (0.4) |
| | Macro F-1 | 74.1 (0.7) | **74.9 (0.7)** | 74.6 (0.4) | 72.9 (0.7) |
| **Text Classification Average Score** | | 76.6 | **76.8** | 76.6 | 76.3 |
| **bc5cdr** | Micro F-1 | 97.0 (0.0) | 97.0 (0.0) | 96.9 (0.0) | **97.2 (0.0)** |
| | Macro F-1 | 88.3 (0.0) | 88.3 (0.1) | 87.9 (0.0) | **88.7 (0.1)** |
| **JNLPBA** | Micro F-1 | 93.4 (0.1) | **93.5 (0.0)** | 93.4 (0.0) | **93.5 (0.0)** |
| | Macro F-1 | 78.0 (0.3) | **78.6 (0.1)** | 77.9 (0.1) | 78.5 (0.1) |
| **NCBI-disease** | Micro F-1 | **98.2 (0.0)** | 98.0 (0.0) | 98.1 (0.0) | **98.2 (0.0)** |
| | Macro F-1 | **91.4 (0.1)** | 90.5 (0.1) | 90.7 (0.1) | 91.3 (0.1) |
| **NER Average Score** | | 91.1 | 91 | 90.8 | **91.2** |
| **TriviaQA** | Micro F-1 | 62.9 (0.1) | 61.4 (0.1) | 59.1 (0.1) | **63.6 (0.1)** |
| | Macro F-1 | 66.2 (0.1) | 64.7 (0.1) | 62.4 (0.1) | **66.8 (0.1)** |
| **SQuAD** | Micro F-1 | 76.6 (0.1) | 76.3 (0.1) | 72.5 (0.1) | **77.1 (0.1)** |
| | Macro F-1 | 85.1 (0.1) | 84.8 (0.0) | 82.3 (0.1) | **85.4 (0.0)** |
| **QA Average Score** | | 72.7 | 71.8 | 69.1 | **73.2** |

Table 12: DistilBERT Results for Reduced Models. **Bold** indicates the best average score between the reduced and fully fine-tuned versions of each model. Each score represents the average score of 10 runs, with the standard errors for each score in parentheses.

| | | MiniLM: 6L-H768 | | | |
|---|---|---|---|---|---|
| | | **2 Layers Reduced** | **3 Layers Reduced** | **4 Layers Reduced** | **Fully Fine-tuned** |
| **ChemProt** | Micro F-1 | **79.4 (0.3)** | 78.3 (0.4) | 79.0 (0.2) | 79.3 (0.3) |
| | Macro F-1 | 51.8 (0.4) | 50.6 (0.4) | 52.0 (0.2) | **52.6 (0.4)** |
| **SciCite** | Micro F-1 | 85.4 (0.1) | 85.8 (0.2) | 85.9 (0.1) | **86.0 (0.2)** |
| | Macro F-1 | 84.1 (0.2) | 84.5 (0.2) | 84.5 (0.1) | **84.6 (0.2)** |
| **SciERC-Rel** | Micro F-1 | 84.7 (0.3) | 83.9 (0.3) | 84.1 (0.4) | **86.3 (0.2)** |
| | Macro F-1 | 75.0 (0.4) | 74.8 (0.4) | 75.3 (0.6) | **78.2 (0.6)** |
| **Text Classification Average Score** | | 76.7 | 76.3 | 76.8 | **77.8** |
| **bc5cdr** | Micro F-1 | 96.1 (0.3) | **96.8 (0.0)** | 96.6 (0.0) | **96.8 (0.2)** |
| | Macro F-1 | 84.6 (1.1) | **87.8 (0.1)** | 86.6 (0.0) | 87.5 (1.0) |
| **JNLPBA** | Micro F-1 | 93.2 (0.0) | 93.2 (0.0) | **93.3 (0.0)** | 93.3 (0.0) |
| | Macro F-1 | **77.5 (0.1)** | 77.3 (0.1) | 77.3 (0.1) | 76.9 (0.2) |
| **NCBI-disease** | Micro F-1 | **98.3 (0.0)** | 98.2 (0.0) | 98.2 (0.0) | **98.3 (0.0)** |
| | Macro F-1 | **92.1 (0.1)** | 91.1 (0.1) | 91.0 (0.1) | **92.1 (0.1)** |
| **NER Average Score** | | 90.3 | 90.7 | 90.5 | **90.8** |
| **TriviaQA** | Micro F-1 | 70.2 (0.1) | 68.9 (0.1) | 65.5 (0.1) | **70.4 (0.2)** |
| | Macro F-1 | 73.4 (0.1) | 72.2 (0.1) | 68.9 (0.1) | **73.8 (0.2)** |
| **SQuAD** | Micro F-1 | 77.6 (0.1) | 75.6 (0.1) | 65.4 (0.2) | **78.9 (0.1)** |
| | Macro F-1 | 86.4 (0.1) | 85.0 (0.1) | 77.0 (0.1) | **87.0 (0.1)** |
| **QA Average Score** | | 76.9 | 75.4 | 69.2 | **77.5** |

Table 13: MiniLM L6-H768 Results for Reduced Models. **Bold** indicates the best average score between the reduced and fully fine-tuned versions of each model. Each score represents the average score of 10 runs, with the standard errors for each score in parentheses.

| | | MiniLM: L6-H384 | | | |
|---|---|---|---|---|---|
| | | **2 Layers Reduced** | **3 Layers Reduced** | **4 Layers Reduced** | **Fully Fine-tuned** |
| **ChemProt** | Micro F-1 | 75.4 (0.5) | **76.9 (0.2)** | 74.9 (0.3) | 74.6 (0.4) |
| | Macro F-1 | 47.3 (0.7) | **50.4 (0.2)** | 48.8 (0.4) | 47.1 (0.8) |
| **SciCite** | Micro F-1 | 84.4 (0.1) | **85.4 (0.1)** | 85.1 (0.1) | 84.4 (0.1) |
| | Macro F-1 | 82.8 (0.1) | **83.7 (0.1)** | 83.4 (0.1) | 82.8 (0.1) |
| **SciERC-Rel** | Micro F-1 | 83.2 (0.3) | 82.6 (0.3) | **83.3 (0.2)** | 79.5 (0.9) |
| | Macro F-1 | 72.7 (0.6) | 72.1 (0.6) | **73.7 (0.3)** | 68.9 (1.1) |
| **Text Classification Average Score** | | 74.3 | **75.2** | 74.9 | 72.9 |
| **bc5cdr** | Micro F-1 | 96.6 (0.0) | 96.3 (0.0) | 95.6 (0.0) | **96.9 (0.0)** |
| | Macro F-1 | 86.9 (0.1) | 85.9 (0.1) | 83.2 (0.1) | **88.3 (0.1)** |
| **JNLPBA** | Micro F-1 | 93.0 (0.0) | 92.2 (0.0) | 92.0 (0.0) | **93.3 (0.0)** |
| | Macro F-1 | 76.3 (0.1) | 74.0 (0.1) | 73.6 (0.1) | **77.2 (0.1)** |
| **NCBI-disease** | Micro F-1 | 98.0 (0.0) | 97.9 (0.0) | 97.7 (0.0) | **98.2 (0.0)** |
| | Macro F-1 | 90.6 (0.1) | 89.9 (0.1) | 88.9 (0.1) | **91.7 (0.1)** |
| **NER Average Score** | | 90.2 | 89.4 | 88.5 | **90.9** |
| **TriviaQA** | Micro F-1 | 66.6 (0.1) | 65.6 (0.1) | 63.4 (0.1) | **67.6 (0.2)** |
| | Macro F-1 | 69.9 (0.1) | 69.2 (0.1) | 67.0 (0.1) | **71.0 (0.2)** |
| **SQuAD** | Micro F-1 | **81.6 (0.0)** | 80.9 (0.1) | 74.2 (0.2) | **81.6 (0.1)** |
| | Macro F-1 | **89.7 (0.0)** | 89.0 (0.0) | 84.5 (0.1) | 89.6 (0.0) |
| **QA Average Score** | | 76.9 | 76.2 | 72.3 | **77.4** |

Table 14: MiniLM L6-H384 Results for Reduced Models. **Bold** indicates the best average score between the reduced and fully fine-tuned versions of each model. Each score represents the average score of 10 runs, with the standard errors for each score in parentheses.

| **Task** | **Averages** | **Standard Recycling** | **Adapter-Based Recycling** |
|---|---|---|---|
| **Classification** | Training Time | 2204 | 2349 |
| | Epochs | 38 | 42 |
| **NER** | Training Time | 4269 | 3857 |
| | Epochs | 43 | 39 |
| **QA** | Training Time | 8252 | 8513 |
| | Epochs | 6 | 7 |

Table 15: Average Training Times and Epochs for Embedding Recycling (seconds for training time, count for epochs). **Standard Recycling** corresponds to layer recycling on a reduced transformer model. **Adapter-Based Recycling** corresponds to layer recycling on a reduced frozen transformer model with added trainable Adapter modules. Training time and epoch averages are the averages across the RoBERTa, BERT, SciBERT, DeBERTa V2 XL, and T5-Large transformer models and the text classification, NER, and QA datasets tested.

