# OpenReview forum: "Embedding Recycling for Language Models"
_TMLR — Rejected by TMLR_

### Review · Reviewer_hmHs · 2022-09-01

**Summary Of Contributions:**

This paper proposes embedding recycling, where the intermediate representations of the data examples are cached (e.g. saved on disk) and frozen, during training or inference the cached embeddings of the corresponding data example are retrieved and the computation starts there – thus the proposed method could save the forward computation at both training and test time. This paper shows that in both full fine-tuning and adapter-tuning cases, it is possible to freeze the first half of the layers with only a slight performance drop.


**Broader Impact Concerns:**

I don't have any concerns on the ethical implications of the work

**Requested Changes:**

I think the work needs significant changes to have my recommendation for acceptance:

1. I would like to see more ablation experiments to prove that the caching and retrieving mechanism is necessary – why not just freeze the first X layers during training?

2. I think the experimental setup about inference-time comparison is inappropriate – related content needs to be re-written.


**Strengths And Weaknesses:**

### Strengths:
1. The proposed method is simple and efficient
2. The results are good
3. The paper is well-written



### Weaknesses:

1. I think it is trivial to just freeze the first X layers of a model during training – for example, it is not new to tune the classification head or the last X layers of a model to save memory. The main difference of this work in my view is that it does not only freeze, but also caches and retrieves the representations as “embedding recycling”, to amortize the forward computations of possibly many epochs with a one-time compute. However, this motivation is not sufficiently strong – in a large dataset there are often few epochs during training, and the proposed method tradeoff the extra storage and the retrieval for forward computation. I would like to see how efficient the method is compared to just freezing the first half of layers during training (without cache and retrieval).


2. Most importantly, I think the proposed method is not applicable at all during inference – where efficiency is often more important than at training time. This paper assumes that the entire test data is known in advance and precomputes their intermediate representations, then they compare the inference speed without counting into the precomputation time. There are two problems with this experiment: 1. The assumption is too strong and unrealistic. In practice the entire test data is not known – for example, the method cannot be deployed for real-time users; 2. Even under this assumption, the comparison is inappropriate – the precomputation time should be counted as well and if so, I think the proposed method would actually increase the latency due to retrieval.

---

> ### Author Response · Authors · 2022-09-16
> **Response to Reviewer hmHs**
>
> We have added a training time comparison to merely freezing the first half of the network (see Table 4). Embedding recycling offers 24-44% faster training compared to just freezing half of the network for our best-performing models. For our compact models, embedding recycling can also accelerate training but more than 12 epochs are needed. However, the distilled models typically require more than 12 epochs to converge so we conclude that embedding recycling is generally better than partially freezing a model during training. Using the early stopping criteria that worked best on our validation set, our training runs tend to take 8-20 epochs per task, creating a large number of redundant forward passes using the freezing strategy (and we note that with our approach, cached embeddings can be reused across tasks as well as across epochs for the same task).
>
> As noted in our general response, we have also performed a measurement that takes the precomputation time into account in Table 4, with further elaboration in section 5.3.  For the setting we are targeting, where we want to perform many runs over the same corpus, the embedding recycling approach offers significant speed-ups for both training and inference.

---

### Review · Reviewer_tBx8 · 2022-09-02

**Summary Of Contributions:**

This paper proposes a method to reduce the fine-tuning and inference costs of pre-trained language models (PLMs, e.g. BERT) through **embedding recycling**, which freezes and caches the hidden layer activations (i.e. the contextual embeddings) at the lower layers of PLMs when fine-tuning on a given task of interest. This speeds up the fine-tuning and inference process by virtue of not having to re-compute the frozen hidden state activations at the lower layers at fine-tuning time, which can simply be retrieved from a cache memory that was constructed at the previous run of the model (before the fine-tuning process began). To further reduce the fine-tuning computation requirements of the model, the embedding recycling technique can be augmented with adapter layers, where only a small number of new parameters are added and modified when fine-tuning the upper layers of the PLM, whereas the rest of the parameters remain frozen.

Experiments on various PLMs (e.g. BERT, SciBERT, RoBERTa, T5, etc.), tasks, and domains (including multiple scientific domains) demonstrate that the proposed approach achieves up to >90% inference & fine-tuning speed-up over the baseline model, with relatively small degradations in model performance, although some tasks (e.g. QA tasks) and models (e.g. T5-Large) exhibit larger degradations than others (classification & NER tasks).

**Broader Impact Concerns:**

No broader impact concerns from my end.

**Requested Changes:**

1. **Critical**: The weaknesses that I raised in points 1-2 above, namely regarding the usability of the approach for real-world applications that often feature new and/or human-crafted inputs on the spot, are fairly fundamental in nature, so it is hard to pinpoint what change can be done to address them. Nevertheless, I would like to hear the authors' reply to this concern, whether during the authors' response period or as an additional discussion section in the paper.

2. **Recommended**: Add a discussion on the relation with prior work on Transformers + memory (see the citations on point 3 of the weaknesses section above).

3. **Recommended**: Add a discussion about the novelty of the proposed approach.

4. **Recommended**: Evaluate the proposed approach on GLUE, and benchmark against various prior work on model compression that was also evaluated on GLUE.

5. **Presentational**: in page 5, it is unclear to me what "immutable nature" means. Also, it would be nice to see the test scores of each model alongside the inference compute required for each model (which are currently in 2 separate tables), so that we can more easily contextualise the trade-off between model performance and inference time. Also, in page 4, "we evaluate whether **the** combining layer recycling ..." -> remove "the".

**Strengths And Weaknesses:**

### Strengths

1. Given the prohibitive costs of deploying PLMs on real-world applications, which can render them inaccessible to researchers and organisations that lack a large number of expensive computational hardwares like multiple GPUs, how we can reduce the inference and fine-tuning costs of these models is an extremely important research question in the field. This paper makes progress in this important open research question, and achieves a strong fine-tuning and inference speed-up, in most cases with only a relatively minor degradation in model performance.

2. The paper is empirically rigorous: It conducts experiments on multiple tasks and domains, including scientific ones, and accounts for the effects of different model random seeds. This is something that we should do more in the field.

3. The proposed approach is simple, intuitive, and easy-to-implement, which can encourage broader adoption within the field. The proposed approach is also general-purpose and model-agnostic, and can be implemented on top of any PLMs (which the paper does by applying the approach on top of BERT, RoBERTa, SciBERT, T5, etc., and demonstrating a fairly consistent result across these different PLMs).

4. The paper is overall clear and well-written.

### Weaknesses
Despite the strengths above, I still have some major concerns with the work in its current form.

1. I am not convinced that the approach will be applicable in real-world applications. A key prerequisite of the speed-up is that the model needs to have "cached" lower-layer hidden state activations for the test/fine-tuning input. While this holds true for many commonly-used datasets like Wikipedia, books, etc., in practice people would like to use the PLMs on **new inputs** that the model often will not have even seen before; hence there would be no corresponding entry on the memory cache. For instance, we would often like to run these PLMs on brand-new tweets, news articles, scientific papers, or user-provided utterances; hence the model is unlikely to have seen these inputs before by virtue of these test inputs' utmost recency. There will be **no speed-up** for these cases when we are doing inference without any fine-tuning, as there will be no cache memory entries corresponding to these utterances and hence the PLMs will need to be run from scratch for these new inputs.

2. Related to point 1 above, it will be difficult to predict in advance what the end users will use the PLMs for. Even if we disregard the very recent test input mentioned in point 1 above, in order to achieve full coverage, the memory cache ideally needs to cover all the English text that is available thus far (in order to avoid re-computing the PLM's lower layers on the new input as much as possible). This would be prohibitively expensive: the cleaned C4 corpus in English has ~800 GB of text, and under the approach, each word position in the dataset would be associated with a hidden state activation of the cached lower layer, which means that the ideal memory cache size may well have a size of many terrabytes. This means that retrieval from the memory cache would also be very slow.

3. There is a lot of past literature on memory and caching prior hidden states within Transformer models, such as Transformer-XLs, memorising Transformers (Yuhuai Wu et al., 2022), and neural cache (Edouard Grave et al., 2016). There is also prior work that investigates whether we need to maintain a memory cache at every Transformer layer, or only at certain Transformer layers (Jack Rae & Ali Razavi, 2020), which is somewhat related to this work, which only needs to save the topmost lower hidden layer into the memory cache. It would be nice to mention the connection with this line of work in the related work section. There is also a lot of prior work on retrieval memory, such as REALM (Kelvin Guu et al., 2020) and DPR (Vladimir Karpukhin et al., 2020).

4. The proposed approach is not particularly novel. The embedding recycling approach basically freezes the lower layers of the PLMs (a known technique that has been explored by Jaejun Lee et al., 2019), stores them into a memory cache (also a known technique in the memory + Transformers literature, see point 3 above for some representative citations), and combines them with other known techniques for reducing fine-tuning & inference computation costs, such as adapter layers and using half-precision floating points (i.e. quantisation). All of these are known techniques that have already been shown to do well in prior work.

5. It would be nice to see results on commonly-used benchmarks like GLUE, where a lot of prior work on model compression has already evaluated on. This would better situate the findings of this work, and further improve the credibility of the findings.

---

> ### Author Response · Authors · 2022-09-16
> **Response to Reviewer tBx8**
>
> Embedding recycling can provide speed-ups for new inputs (e.g. news articles, tweets) provided we wish to run multiple distinct models on those inputs.  How the savings scales with the number of models to be applied is now reported in Table 4 and section 5.3.  Our work is targeted at the case where we want to apply many models to the same text, and new models and tasks may arise over time.
>
> As far as the size of the cache being prohibitive, if we process huge corpora like C4, we would advocate for only caching activations for text in cases where many models will be run over that text.  In those settings, the time savings will often exceed the cache storage costs as we discuss in our efficiency analysis in Section 5.3.
>
> Thank you for the pointers to previous work.  We have added discussion of those papers to our related work section.
>
> Regarding novelty, we agree that our technique essentially combines two existing ideas (caching transformer state and freezing layers).  Part of the reason we targeted the TMLR journal is because of its editorial policy which requires only that submissions be accurate and of interest to at least some of the journal’s audience— “even if the level of contribution is modest.”  We do not claim radical methodological novelty, but we believe that our experiments present a new, helpful, and somewhat surprising demonstration of the utility of embedding recycling in practice.
>
> By “immutable,” we just mean that published scientific papers tend to remain frozen in their published state, the text does not change.  This means that the same cached embeddings for a paper can be reused for any subsequent models.
>
> Thank you for the suggestion to look at GLUE tasks.  Previous work has already measured the accuracy of half-frozen models on some GLUE tasks (the prior work does not consider caching, but the accuracy it reports should be comparable to our approach).  The work by Lee et al. (2019) shows that half-frozen models of the size of BERT-base and BERT-large achieve relatively high accuracy on four GLUE tasks (see [R3] Figure 1).  We added experiments analyzing whether this trend continues with our larger best-performing model (DeBERTa V2 XL) in the current revision. We were able to complete those runs for six GLUE tasks, as reported in section 6.2: CoLA, SST-2, MRPC, STS-B, MNLI, and QNLI.  We find that embedding recycling is successful, with a small loss in F1 (0.2-0.5 points) in exchange for significant speedups in training and inference as outlined in Tables 3 and 4.  We will add experimental results for the remaining GLUE tasks and additional models in our final version of the paper.
>
> Thank you for the presentational suggestion on showing the test scores alongside the inference computations required for each model. To help contextualize the tradeoff between model performance and inference speedups, we added a column in Table 3, which indicates the average F1 loss using embedding recycling for each model across our tasks.
>
> [R3] Lee, Jaejun, Raphael Tang, and Jimmy Lin. "What would elsa do? freezing layers during transformer fine-tuning." arXiv preprint arXiv:1911.03090 (2019).

---

> > ### Comment · Reviewer_tBx8 · 2022-09-27
> > **Response to the Authors' Response**
> >
> > Thank you for the authors' response! I appreciate the fact that the paper clarified in the paper regarding when exactly the embedding recycling approach has efficiency benefits; I also appreciate the encouraging results on GLUE.
> >
> > Regarding the first two points that I raised, the authors' response clarified that it would be useful to cache the contextual embeddings when multiple predictions need to be made from the same document. This would represent one example use case, but I am still unsure whether or not this represents the majority of real-world NLP use cases. For instance: (i) How often are NLP systems deployed on the same, frequently used documents? (ii) How can we predict in advance which documents should be cached, and which documents should not? (iii) Won't a substantial part of NLP use cases require making predictions on brand new articles (including, e.g. the latest ArXiv papers, or the latest news), or on user-generated inputs --- both of which are unlikely to have been seen and cached before?
> >
> > In my view, having some empirical data on what proportion of real-world NLP use cases would correspond to the case where embedding recycling is helpful would be beneficial to strengthen the paper.

---

> > > ### Author Response · Authors · 2022-09-30
> > > **Response to Reviewer tBx8’s reply**
> > >
> > > Thank you for the additional suggestions and insights.  We agree that it is not yet known whether a majority of workloads will benefit from embedding recycling (ER).  We appreciate the suggestion that we try to quantify the proportion of cases where ER helps.  While we do not yet have such a quantitative estimate, below we provide some data and use cases suggesting that for multiple important workloads, ER can provide benefits.  We will expand on this discussion in our revisions.
> > >
> > > Consider a scholarly research tool that operates on a corpus of scientific papers, providing users with features such as extracted paper keywords, entities, summaries, recommendations, and so on.  Even though many new scientific papers are published each year (and the rate is increasing), the fraction of new papers remains a very small proportion of all papers.  According to one estimate, the number of new papers published in 2021 was only 6.3% of that published in the 25 years prior [1].  We expect that any actively-developed research tool using this corpus will need to reprocess the corpus frequently relative to this growth rate, either to add new models that power new features or to update models with better training data or optimization techniques.  At the current rate of growth, even if the corpus were only reprocessed once every 11 years, more than half of the documents would be in the cache.
> > >
> > > Another more general class of ER use cases arises in the context of data cascades [2,3]—compounding events in ML systems that cause negative downstream effects based on data processing. For high-stakes AI models addressing challenges like healthcare or conservation, it is especially crucial to filter and process data to ensure quality for downstream tasks. Within this scenario, ML practitioners are frequently reprocessing the same data as they iteratively construct new ML systems, making it an appealing use case for embedding recycling.
> > >
> > > Regarding predicting which documents will be reprocessed, we believe that practitioners generally have the necessary domain expertise and intuition to know when it makes sense to cache a corpus.  For example, in the first scenario above, the developers of the scholarly research tool can estimate the rate at which their corpus grows and the frequency with which it needs to be reprocessed based on past usage and their future plans (e.g., for adding new features or upgrading models). In such cases, predicting what to cache is not a problem.  In other cases where prediction is more difficult or where the space to store the cache is limited, standard cache replacement policies could be tried.  We note that in our use case where whole documents of context-sensitive embeddings are cached, querying the cache and missing adds negligible time cost–so the storage cost of the cache would be the limiting factor.  In a local computing environment, that storage cost is small relative to the cost of reprocessing (although it is more significant in a cloud computing environment), but as the exact cost depends on specific use case assumptions we will include a more detailed analysis in our revisions.
> > >
> > > Finally, we note that embedding recycling actually can be applied to new inputs if we know in advance that there are multiple tasks we wish to perform (e.g., for a scientific paper, we might want to both classify its field of study and extract its named entities).  In such cases, we can immediately cache representations to share across all tasks and save time overall.  These scenarios arise in many industry settings, such as virtual assistants, where a product needs to support multiple features for each input.
> > >
> > > [1] https://www.scimagojr.com/countryrank.php?order=itp&ord=desc
> > >
> > > [2] https://ai.googleblog.com/2021/06/data-cascades-in-machine-learning.html
> > >
> > > [3] https://storage.googleapis.com/pub-tools-public-publication-data/pdf/0d556e45afc54afeb2eb6b51a9bc1827b9961ff4.pdf

---

### Review · Reviewer_TpDu · 2022-09-06

**Summary Of Contributions:**

Paper proposes a method for training and inference time efficiency of pretrained large language models. The main motivation is that the text being processed has already been processed through a model before, and the activations from previous runs may be re-used to speed up the latter runs. In principle proposed method relies on reusing activations from previous model runs during training and inference.

Author test various options by caching intermediate layers' outputs from a pretrained model and fine-tuning the remaining layers for new tasks. In particular they test two layer (activation) recycling approaches: one that uses standard fine-tuning and another that uses parameter-efficient adapters. Cross-model reuse also tested without much success.

Experiments on multiple pretrained models and multiple tasks show 2x speedup during training and close to 2x speedup for inference while having small impact on accuracy.

The latency analysis and discussion / future work sections add value to the paper.

**Requested Changes:**

- Please put Table 1 and 2 on the pages where they are referenced.

**Strengths And Weaknesses:**

Strengths:
- Efficiency in NLP has been an important topic lately, and this paper explores a wide area in that space -- expected to be useful for efficiency research, given that the authors promise that they'll be releasing the code.
- Raises interesting questions especially about the behavior of classification, QA and generative task responses of the proposed method. One would expect the authors have a follow up paper analyzing these topics in details.
- Detailing the latency analysis on different hardwares might be handy for downstream users.

Weaknesses:
- My main concern is about the proposed inference efficiency setup. How can we recycle embedding at inference time if the example has never been seen before? And if the example has seen before, how that should not be counted as test set pollution of the training corpora? (at directions for future work you indicate that the examples are seen, hence I'm assuming the latter being true)
- Not particularly ground-breaking, but important research and findings as it adds value to the Efficiency in NLP space, therefore I'm not counting this as a major weakness of the paper.

---

> ### Author Response · Authors · 2022-09-16
> **Response to Reviewer TpDu**
>
> Thank you for your helpful comments.  Regarding the inference-time speedup, we assume that the input text examples have been seen previously, but we do not assume to have seen the ground-truth labels (hence, there is no test set pollution of the training set).  See the general response for more discussion on the inference-time speedup issue.
>
> We also moved Tables 1 and 2 closer to the pages where they are referenced. Please let us know if the formatting is still unclear or confusing.
>
> Thank you for the recommendation to analyze the differences across tasks more thoroughly, we agree this is an important direction for future work.

---

### Author Response · Authors · 2022-09-16
**General Response to Reviewers**

Dear AE and all reviewers,

We sincerely appreciate the time and effort you all put into reviewing our paper. We also thank you for your analysis of our methods and results, which helped us strengthen our paper.

All three reviewers ask whether our technique is applicable at inference time.  We appreciate the opportunity to clarify that our approach is applicable at inference time. In fact, using embedding recycling to speed up inference in a practical setting was the original motivation for our work.  Our approach provides savings when we want to run inference many times over the same text.  Consider a scientific search engine, which might want to run many NLP models over every paper (e.g. entity linking, summarization, definition detection, citation extraction and classification, relation extraction, topic classification, recommendation, etc.) and is frequently developing new improved models for each of these tasks.  Under current practices, each one of these potentially hundreds of inference runs would require executing a costly pretrained transformer over the whole corpus, which is prohibitively expensive.  Previous work has pointed out this expense in other domains: for example, Du and Ott et al. (2020) note how for a news article, we may want to predict various characteristics such as topic, sentiment, text quality, humor, aggression, and more [R1].  Likewise, Wei et al. (2022) discuss how a commercial AI assistant performs multiple classifications for a single user query, such as emotion recognition, incoherence detection, domain classification, intent classification, named entity recognition, slot filling, etc. [R2].

Our approach tries to reduce this expense by paying a one-time cost to produce a cached embedding layer, and then reuses that layer for all future model runs over the same text, making them less expensive.

We have clarified this motivation in the revised paper in two ways.  First, we have expanded our discussion of the target use case in the introduction.  We also explicitly note that our approach does not provide savings the very first time we run a model on a text.  Secondly, we have now explicitly accounted for the one-time cost of producing the cached embeddings as requested by reviewer tBx8. Table 4 now shows how during training, saving the embeddings on the first pass over the training data has only a small impact on the total speedup provided by embedding recycling.  Also, as we note in Section 5.3, for inference we report the speedup assuming that the cache is already precomputed.  The total amortized speedup including the time to compute the cache will depend on the total number of inference passes we must perform for new models and tasks being run over the same text.  We believe that dozens of passes are a practically relevant setting for many real-world applications, at which point the total time savings will be approximately equal to (i.e., within a few percent of) that reported in our result tables.

Finally, one additional important clarification that was missing from our original submission: for the T5 results, we consider caching only intermediate layers of the encoder, not the decoder.  This means that the efficiency improvements for T5 will generally be smaller than (about half of) the gains of the other models we evaluate.  We have clarified this in the revised paper.

We also respond to questions and concerns of individual reviewers directly in the comments for each review.

Thank you again to all the reviewers and AE for your time and effort. If you have any additional comments or questions, please let us know.

Regards,

Authors

[R1] Du, Jingfei, et al. "General purpose text embeddings from pre-trained language models for scalable inference." arXiv preprint arXiv:2004.14287 (2020).

[R2] Wei, Tianwen, Jianwei Qi, and Shenghuan He. "A Flexible Multi-Task Model for BERT Serving." arXiv preprint arXiv:2107.05377 (2021).

---

### Decision · Action_Editors · 2022-10-16

**Recommendation:** Reject

**Comment:**

I would first like to thank the authors and the reviewers for their meaningful exchange and for their efforts in the rebuttal phase.

Starting from the test scenario were multiple tasks need to be performed on the same document, the authors propose to "cache" the representations of the first layers of a pre-trained backbone and measure the speed-up offered by caching and then fine-tuning only the last layers rather than, i.e., by "freezing" the representations of the first layers.

Reviewers were mainly concerned about the following points:

1. Applicability of the method at inference time: what to do for a new test input?
2. Limited novelty: freezing intermediate layers has already been studied in the literature
3. Some missing related work
4. Some more well studied tasks such as GLUE: included in the updated version

The authors wrote in the rebuttal that:

1. There use cases where the same document might undergo different tasks (QA, NER, etc.) In this case, the speed-ups would be obtained by caching the representation of the document.
3. Added related works in the revised version as asked by reviewer tBx8
4. Added results on GLUE as demanded by reviewer tBx8

After the rebuttal, two reviewers are leaning reject and one reviewer decided to give accept. Negative reviewers were mostly concerned about the limited use case application of the methods, where the speed-up would be obtain only when the same document seen during training must be re-processed for a task.

I re-read the paper and your exchanges carefully. Personally, I think the contribution in this paper is too thin, even for TMLR, for which novelty shouldn't be a factor in acceptance or reject decisions. Leaving out the fact that the application scenario is a bit "niche" (multiple tasks on the same document), there is a lot of work in the literature already examining the impact of freezing some layers and retraining only upper layers (e.g. Wei et al. 2022, cited in the paper). Overall, the novelty of this work is rather small when considered in the context of previous works.

Even if novelty is not a factor which determines the acceptance of a paper for TMLR, the interest of the paper to the TMLR audience is, and I believe that the limited novelty and the limited number of scenarios in which the technique can be applied might affect the interest the audience might find in this paper. One suggestion I could offer to the authors is to focus the study more on the "transferability" of the cached representations across models, which is now only addressed in small details in the current draft. That might offer a more novel perspective on these problems.

**Audience:**

The limited novelty and the limited number of test scenarios in which the technique can be applied might affect the interest the audience might find in this paper.

**Claims And Evidence:**

The claims made in the submission are supported by accurate and clear evidence.